# Underwater Object Detection Using TC-YOLO with Attention Mechanisms

**DOI:** 10.3390/s23052567

**Published:** 2023-02-25

**Authors:** Kun Liu, Lei Peng, Shanran Tang

**Affiliations:** School of Civil Engineering and Transportation, South China University of Technology, Guangzhou 510641, China

**Keywords:** object detection, underwater image, YOLOv5, coordinate attention, transformer

## Abstract

Underwater object detection is a key technology in the development of intelligent underwater vehicles. Object detection faces unique challenges in underwater applications: blurry underwater images; small and dense targets; and limited computational capacity available on the deployed platforms. To improve the performance of underwater object detection, we proposed a new object detection approach that combines a new detection neural network called TC-YOLO, an image enhancement technique using an adaptive histogram equalization algorithm, and the optimal transport scheme for label assignment. The proposed TC-YOLO network was developed based on YOLOv5s. Transformer self-attention and coordinate attention were adopted in the backbone and neck of the new network, respectively, to enhance feature extraction for underwater objects. The application of optimal transport label assignment enables a significant reduction in the number of fuzzy boxes and improves the utilization of training data. Our tests using the RUIE2020 dataset and ablation experiments demonstrate that the proposed approach performs better than the original YOLOv5s and other similar networks for underwater object detection tasks; moreover, the size and computational cost of the proposed model remain small for underwater mobile applications.

## 1. Introduction

Ocean exploration and exploitation have become commanding heights of the economy in many countries. Underwater object detection enables intelligent underwater vehicles to locate, identify, and classify underwater targets for various tasks, which is a vital sensing technology that has extensive applications in ocean exploration and salvage, offshore engineering, military operations, fishery, etc. [1]. Compared with sonar detection [2], cameras can obtain close-range information so that intelligent underwater vehicles can better perceive the surrounding environment. Image-based underwater object detection has been developed rapidly in recent years, along with the development and application of deep learning in computer vision. However, compared with other typical computer vision tasks, underwater object detection presents unique challenges, including poor image quality, small and dense targets difficult to detect, and limited computation power available within underwater vehicles.

Image enhancement is an effective method to improve underwater image quality and thus improve the accuracy of underwater object detection [3]. Both traditional image enhancement algorithms, such as multi-scale Retinex with color restoration [4] and defog [5], and deep learning algorithms, such as generative adversarial network (GAN) [6], have been applied to improve image quality. Particularly for underwater object detection, several researchers applied image enhancement algorithms based on the Retinex theory and obtained clearer underwater images, but the final prediction results were not significantly improved because only enhancing underwater images does not guarantee better prediction results [7,8]. GAN was used for color correction in underwater object detection tasks [9], but the resulting detection network is rather large and requires costly computation. The image enhancement technique employed should be computationally efficient and combined with other approaches to improve the overall performance of underwater object detection.

Targets of underwater object detection are often small and dense. Deep convolutional neural networks (CNNs) enable multi-layer non-linear transformations that effectively extract the underlying features into more abstract higher-level representations, allowing effective detection when there is target occlusion or the target size is small. YOLO (You Only Look Once) is a series of widely-used CNNs for object detection tasks [10,11,12,13,14,15]. Sung et al. proposed a YOLO-based CNN to detect fish using real-time video images and achieved a classification accuracy of 93% [16]. Pedersen et al. adopted YOLOv2 and YOLOv3 for marine-animal detection [17]. Other researchers applied attention mechanisms to detection networks for better identification of small and dense targets [18,19,20].

Applications of attention mechanisms have been proven effective for object detection; nevertheless, most existing research improves detection performance at the considerable expense of computational cost, as the demand for attention computation is high. Computational capacity and power supply available to underwater vehicles are usually very limited. Increasing computational demand for object detection would result in the need for underwater vehicles to be connected with cables for data transfer or power supply, dramatically limiting the operating range and the level of autonomy of underwater vehicles. Therefore, attention mechanisms should be carefully integrated with detection networks, such that the increases in model size and computational cost are minimized for underwater applications.

In this study, we propose a new underwater object detection approach that has three major improvements over the original YOLOv5. A new detection network named TC-YOLO was developed to integrate Transformer self-attention and coordinate attention mechanisms for better small object identification. An image enhancement algorithm, contrast-limited adaptive histogram equalization, was used to improve the underwater image quality. Optimal transport label assignment was used to replace the original YOLOv5 label assignment scheme for network training. Experiments on the Real-world Underwater Image Enhancement (RUIE2020) dataset and ablation experiments were carried out. The results demonstrated the effectiveness and superior performance of the proposed approach. Compared with YOLOv5s and other advanced detection networks, the proposed TC-YOLO not only improves the detection performance of underwater objects but also remains relatively small without a significant impact on computational cost.

The remainder of this paper is organized as follows: The previous works related to the proposed approach are briefly introduced in Section 2. The development and details of the proposed approach are presented in Section 3. The proposed method is tested and compared with other detection networks in Section 4, including ablation experiments. Conclusions are finally given in Section 5.

## 2. Related Work

### 2.1. YOLOv5

In order to achieve real-time underwater object detection, many researchers have chosen the YOLO series as the basis for further development. Wang et al. [21] reduced the network model size of YOLOv3 and replaced batch normalization with instance normalization in some early layers, thus enabling underwater deployment while improving detection accuracy. Al Muksit et al. [22] developed the YOLO-Fish network by modifying the upsampling steps and adding a spatial pyramid pool in YOLOv3 to reduce the false detection of small fishes and to increase detection ability in realistic environments. Zhao et al. [23] and Hu et al. [24] improved the detection accuracy by optimizing the network connection structure of YOLOv4 and updating the original backbone network.

YOLOv5, like other YOLO series, is a one-stage object detection algorithm. YOLOv5 includes four variants, namely YOLOv5s, YOLOv5m, YOLOv5l, and YOLOv5x, whose network size and number of parameters increase successively. Figure 1 shows the overall structure of the YOLOv5 network. An image is first processed by the backbone for feature extraction, followed by the neck for feature fusion, and finally is outputted as the head for the prediction of objects. The backbone is used to extract features from images, containing a total of 53 convolutional layers for various sizes of image features. The CSPDarknet53 structure proposed in YOLOv4 is slightly modified and continuously employed in YOLOv5 [25]. The spatial pyramid pooling module in YOLOv4 is replaced by the spatial pyramid pooling-fast module to improve computational efficiency. The neck is used to reprocess the extracted features for various spatial scales, which consists of top-down and bottom-up paths to form a cross-stage partial path aggregation network [26]. The feature pyramid network [27] is used to fuse the features from the top to the bottom, and a bottom-up path augmentation is used to shorten the path of low-level features. The fused features are integrated into the head structure, and three prediction paths are used. Each path fuses the low-level features of different receptive fields and finally outputs the bounding box, confidence, and category of the detected objects.

Mosaic, mix-up, copy–paste, and several other methods are employed in YOLOv5 for data augmentation [10]. YOLOv5 uses complete intersection over union (IoU) loss [28] to compute the bounding box regression loss, in which the distance between the center points of the bounding box and ground truth (GT) and the aspect ratio of the bounding box are rigorously considered. Similar to YOLOv4, the basic anchor size is computed in YOLOv5 using the k-mean algorithm, but YOLOv5 introduces a priori judgment called auto-anchor to enhance the versatility of anchor boxes. If the preset anchor size matches well with a dataset, the recalculation of anchor boxes can be avoided. YOLOv5 inherits the non-maximum suppression method from its predecessors [29], in which two bounding boxes are considered to belong to the same object if the IoU of two bounding boxes is higher than a certain threshold.

The proposed TC-YOLO network was developed based on YOLOv5s. The overall structure of YOLOv5 and the data augmentation methods were preserved, while self-attention and coordinate attention mechanisms were integrated with the backbone and the neck, respectively.

### 2.2. Attention Mechanism

Most of the targets for underwater detection are small and dense, so researchers have introduced attention mechanisms to improve the detection performance. Sun et al. [30] attempted to design a new network using Swin Transformer as the backbone for underwater object detection and obtained similar performance as the Cascade R-CNN with the ResNeXt101 backbone. Other researchers chose to combine attention mechanisms with existing networks to improve the detection accuracy for underwater targets [31,32]. For example, Wei et al. [20] added squeeze-and-excitation modules after the deep convolution layer in the YOLOv3 model to learn the relationship between channels and enhance the semantic information of deep features.

The attention mechanisms used in computer vision are traditionally divided into three types: spatial attention, channel attention, and hybrid attention. Applying the attention mechanism in the spatial domain gives neural networks the ability to actively transform spatial feature maps, for example, the spatial transformer networks [33]. Applying the attention mechanism in the channel domain enables strengthening or suppressing the importance of a channel by changing the weight of this channel, for example, the squeeze-and-excitation networks [34]. Later works, such as the convolutional block attention module (CBAM), employ a hybrid mechanism that combines spatial attention and channel attention modules together and is widely used in convolutional network architectures [35].

Coordinate attention (CA) is another hybrid attention mechanism recently proposed in 2021 [36]. Improved from channel attention, CA factorizes channel attention into two 1-dimensional feature-encoding processes, each of which aggregates features along one of the two spatial coordinates. It encodes both channel correlations and long-range dependencies with precise positional information in two steps: coordinate information embedding and coordinate attention generation. Compared with the CBAM, which computes spatial attention using convolutions, the CA can preserve long-range dependencies that are critical to vision tasks. Additionally, the CA avoids expansive convolution computations, improving its efficiency compared with other hybrid attention mechanisms and enabling its application to mobile networks.

In addition to the above attention mechanisms commonly used in computer vision, Transformer was introduced in 2017 for natural language processing [37], and since then, it has been successfully applied in different neural network architectures for various tasks. Convolutional networks have the problem of limited perceptual fields. Multi-layer stacking is required to obtain global information, but as the number of layers increases, the amount of information is reduced, such that the attention of extracted features is concentrated in certain regions. Transformer, on the other hand, employs the self-attention mechanism that can effectively obtain global information. Furthermore, the multi-head structure used in the Transformer allows better fusion and more expressive capability, in which feature maps can be fused in multiple spatial scales. The Transformer mechanism has been applied to computer vision, such as Vision Transformer [38] and Swin Transformer [39]. However, Transformer modules require a large amount of computation, so they are mostly adopted in large networks and hardly used in mobile networks.

Both Transformer and CA modules were employed in the proposed TC-YOLO network. Transformer is computationally expensive, especially applied to computer vision tasks, such that it is not suitable to completely replace CNN with Transformer for underwater detection applications. Therefore, we embedded a Transformer encoder module into the end of YOLOv5’s backbone to improve the global representation ability of the network. CA can capture not only cross-channel correlation but also orientation- and location-sensitive information, which helps the network to locate and identify targets accurately. In addition, the CA module is flexible and lightweight. Thus, we integrated CA modules into the neck to improve detection performance.

### 2.3. Label Assignment

Label assignment aims to determine positive and negative samples for object detection. Unlike labeling for image classification, label assignment in object detection is not well defined due to the variation in bounding boxes. Zheng et al. [40] proposed a loss-aware label assignment for a one-stage detector for dense pedestrian detection. Xu et al. [41] proposed a Gaussian receptive field-based label assignment strategy to replace IoU-based or center-sampling methods for small object detection. Optimal transport assignment (OTA) was proposed in 2021 for object detection [42]. OTA is an optimization-theory-based assignment strategy, in which each GT is considered a label supplier, and its anchors are regarded as label demanders. The transportation cost between each supplier–demander pair is defined by the weighted summation of its classification and regression losses, and then the label assignment is formulated into an optimal transport problem, which aims to transport labels from GT to anchors at minimal transportation cost.

In YOLOv5, the ratios in height and width between a GT and an anchor box are computed as $r_{h}$ and $r_{w}$, and then $\max(r_{h},r_{w},1/r_{h},1/r_{w})$ is compared with a threshold value of 4. If the maximum is smaller than 4, a positive sample is built for this GT; otherwise, the sample is assigned as negative. This label assignment mainly considers the difference in the aspect ratio between the GT and its anchor boxes, which is a static assignment strategy. Improved from the earlier versions of YOLO, a GT can be assigned to multiple anchors in different detection layers in YOLOv5. However, YOLOv5’s label assignment is still not sufficient for underwater object detection because the same anchor may be assigned multiple times, resulting in missed detection. To deal with dense object detection, we employed the OTA scheme to obtain the label assignment that is globally optimal.

### 2.4. Underwater Images Enhancement

The quality of underwater images is often poor due to harsh underwater conditions, such as light scattering and absorption, water impurities, artificial illumination, etc. Enhancement and restoration techniques are applied to underwater images to improve the performance of underwater object detection. Li et al. [43] proposed an underwater image enhancement framework consisting of an adaptive color restoration module and a haze-line-based dehazing module, which can restore color and remove haze simultaneously. Li et al. [44] developed a systematic underwater image enhancement method, including an underwater image dehazing algorithm based on the principle of minimum information loss and a contrast enhancement algorithm based on the histogram distribution prior. Han et al. [45] combined the max-RGB method and the shades-of-gray method to achieve underwater image enhancement and then proposed a CNN network to deal with low-light conditions. There are other methods used for underwater image enhancement, including spatial domain methods, transform domain methods, and deep learning methods, among which spatial domain methods are considered the most effective and computationally efficient [46].

Several spatial domain image enhancement methods are introduced below. The histogram equalization (HE) algorithm computes the grayscale distribution of an image, stretches its histogram evenly across the image, and therefore improves the image contrast for better separation between foreground and background [47]. However, the HE algorithm increases the sparsity of grayscale distribution and may cause a loss of detailed image-related information. Adaptive histogram equalization (AHE) improves the local contrast and enhances edge details by redistributing the local grayscale multiple times, but it has the disadvantage of amplifying image noise [48]. On the basis of AHE, contrast-limited adaptive histogram equalization (CLAHE) imposes constraints on the local contrast of the image to avoid excessively amplifying image noise in the process of enhancing contrast [49]. Computations of field histograms and the corresponding transformation functions for each pixel are very expensive. Therefore, CLAHE employs an interpolation scheme to improve efficiency at the expense of a slight loss of enhancement quality. Histogram equalization methods only solve the problem of image brightness without adjusting image colors. The Retinex theory can be applied for image enhancement to achieve a balance in three aspects: dynamic range compression, edge enhancement, and color constancy. There are several image enhancement methods developed based on the Retinex theory [50], such as single-scale Retinex, multi-scale Retinex, multi-scale Retinex with color restoration (MSRCR), etc. Retinex-based methods can perform adaptive enhancement for various types of images, including underwater images, but the computational cost is much higher than histogram equalization methods.

To select the most effective image enhancement method for the proposed approach, we tested the processing speed and compared the enhancement results of HE, AHE, CLAHE, and MSRCR, the results of which are presented in Section 3.4.

## 3. Proposed Approach

The following three methods were proposed to improve the performance of a modified YOLOv5 network for underwater object detection:Attention mechanisms were integrated into YOLOv5 by adding Transformer and CA modules to develop a new network named TC-YOLO;OTA was used to improve label assignment in training for object detection;A CLAHE algorithm was employed for underwater image enhancement.

### 3.1. Dataset

We chose to train and test the proposed approach using the RUIE2020 dataset from the Underwater Robot Picking Contest [51]. The images of this dataset present various underwater conditions and offer a comprehensive picture of the underwater environment. The RUIE2020 dataset has four object categories: holothurian, echinus, scallop, and starfish. There are 5543 images in total, and we randomly split the dataset into a training set of 4434 images and a testing set of 1109 images. The RUIE2020 dataset provides box-level annotations with more than 30,000 labels. Sea urchins have the largest number of labels in the dataset, accounting for more than half the labels. In addition, most of the targets in the dataset are small; 90% of the target boxes have an area that is less than 5% of the image area.

### 3.2. TC-YOLO

The modified object detection network is called TC-YOLO, as it includes one Transformer module in the backbone and three CA modules in the neck. Transformer and CA modules were combined with a cross-stage partial (CSP) structure to establish the attention blocks named CSP-TR and CSP-CA, respectively. Similar to most detection algorithms, the detection head was placed after the neck. The network generated feature maps at three scales as the input to the detection head. The feature map at each scale corresponded to three anchors, so in total, nine anchors were obtained by clustering the dataset. The placements of these attention blocks and the overall structure of TC-YOLO are shown in Figure 2.

#### 3.2.1. Transformer Module

In this study, we adopted a modified Transformer encoder module and combined it with the backbone network by placing the module after the minimum feature map to reduce its computational cost (see CSP-TR in Figure 2). Different from other studies in which the Transformer module is used directly, we nested the encoder module inside the CSP structure, which reduced the number of channels of the input feature by half without any loss of local information. The detailed structure of the proposed CSP-TR block is shown in Figure 3. The input of the CSP-TR block is a C × W × H feature tensor, where C is the number of channels; W is the feature width, and H is the feature height. The CSP-TR block ess embedded at the end of the backbone, so the input feature had less width and height and therefore fewer dimensions after flattening. The flattened feature was defined as the patch, and position encoding was carried out by passing patches through a fully connected layer. Position-encoding data and the original patch were then added as the input of the Transformer encoder. We deleted the original linear normalization layer of the encoder and used only one Transformer layer without stacking to minimize the model size. The output of the Transformer encoder remained the same size as the input, so the output patch could be reshaped back to a feature tensor. The resulting feature was concatenated with the feature from the other branch in the CSP structure to recover the original number of channels.

#### 3.2.2. Coordinate Attention

The CA modules were embedded in the neck before the prediction head in this study (see CSP-CA in Figure 2). The arrangement of the proposed method is different from other studies in which convolution-based attention mechanisms are placed in the backbone, and there are two reasons for the proposed arrangement of the CA modules: we already applied the self-attention mechanism in the backbone for feature extraction by replacing a standard CSP block with a CSP-TR block; placing the CA modules before prediction can effectively summarize the global information for different size features after extraction and fusion. Similar to the CSP-TR block, we nested a modified CA module inside the CSP structure to reduce computational costs and preserve local information. The detailed structure of the proposed CSP-CA block is shown in Figure 4. The input of CSP-CA was a C × W × H feature tensor from the end of the network. Coordinate attention was generated by first pooling the X and Y coordinates of the features equally, followed by encoding the pooled features through convolution and normalization, and finally multiplying the encoded features with the original features using the “Reweight” operation. To smooth the activation curve and avoid the gradient problem, we modified the CA module by replacing the original Hard–Swish activation function with the SiLU function, which formed a standard CBS structure in the middle of the CA module. The output of the CA module shared the same size as its input, which was subsequently concatenated with the feature from the other branch of the CSP structure to restore the original number of channels.

### 3.3. Optimal Transport Assignment

The optimal transport label assignment was employed in the proposed approach. The output tensor of the detection network for each feature scale included the distribution of bounding boxes for different target classes within every grid. All the output tensors and the GTs were passed into the OTA module for the computation of classification and regression losses to establish a transportation cost matrix. The optimal transport of labels was obtained by minimizing the cost matrix via the Sinkhorn–Knopp iteration. Based on the optimal transport of labels, the top ten anchors that received the most labels from a GT were selected as the positive samples of this GT. If multiple GTs shared any anchor, further filtering would be required using the cost matrix: The anchor would only assign to the GT that had the smallest cost value with the anchor. We implemented OTA by replacing the original label assignment codes in YOLOv5 with the OTA source codes available from GitHub.

### 3.4. Underwater Image Enhancement

In order to select the most suitable image enhancement algorithm for the proposed approach, we compared the image enhancement results and processing speed of four different algorithms: HE, AHE, CLAHE, and MSRCR. These algorithms can be implemented directly using the OpenCV library. In the implementation of CLAHE, ’clipLimit’ is an important parameter that sets the threshold for contrast limiting to prevent over-saturation in homogeneous areas. These areas are characterized by a high peak in the histogram because most pixels fall in a narrow range of grayscale. The value of ’clipLimit’ is 40 by default but was changed to 2 in this study to avoid color bias. Figure 5 shows five underwater images and their processed images for comparison. It can be seen that the CLAHE algorithm provides a significant improvement over the HE and AHE algorithms, preserving dark and bright details and having less noise interference. The MSRCR algorithm not only improves the details but also restores the original colors, yielding the best overall enhancement results.

We further tested the four different image enhancement algorithms using the original YOLOv5s network with the RUIE2020 dataset. Notably, 80% of the images were randomly sampled as the training set, and the remaining 20% were used as the testing set for validation. The images in the training and testing sets were enhanced using the four algorithms and used to train and test the four YOLOv5s networks. The original images without enhancement were also employed to set a benchmark. All the images in the dataset were scaled to 640 × 640 before being processed, and all tests were performed using the same workstation. The test results of each detection network, including precision, recall, and image processing time, are compared in Table 1. The processing time presented here refers to the time of processing a single image, excluding the inference time of the detection network. YOLOv5s achieved 79.7% precision and 71.1% recall on the original images. Applications of HE and AHE reduced the performance of object detection due to overexposure and stitching issues. The application of CLAHE could effectively reduce the case of missed detection, but its enhancement of dark details may lead to some incorrect detection, causing a slight drop in precision. The MSRCR algorithm led to the best image enhancement result; however, its image processing time was dramatically greater than the other three algorithms. Therefore, we found CLAHE to be the most suitable for real-time underwater object detection tasks.

## 4. Experiments and Results

Experiments were carried out to analyze and verify the proposed underwater detection approach. The proposed TC-YOLO network was developed using the Torch library. All the experiments were performed using a workstation that has 32 GB of RAM and an NVIDIA GTX 1660 GPU with CUDA 10.1 GPU acceleration library. Online data argumentation, such as mosaic and scaling, was activated to prevent over-fitting and ensure the generalizability of the model [13]. All the images were scaled to 416 × 416 for training since the size of input for a mobile network is usually small. For stable batch normalization and further prevention of over-fitting, we set the batch size to 32 and applied an early stop method by training the networks with stochastic gradient descent for 50 epochs [52]. The learning rate and the momentum were set to 0.001 and 0.9, respectively. The warm-up period was set to 3000 iterations, the weight decay was set to 0.0005, and the confidence threshold was set to 0.25 for comparison [53].

### 4.1. Evaluation Metrics

The performance of the detection network was evaluated using the following metrics [54], where *TP* is true positive; *FP* is false positive, and *FN* is false negative:Precision, defined as *TP*/(*TP* + *FP*), reflects the false-detection rate of a network;Recall, defined as *TP*/(*TP* + *FN*), reflects the missed-detection rate of a network;mAP^IoU=0.5^, defined as the mean average precision (mAP) evaluated for all object classes of the entire dataset, in which IoU = 0.5 was used as the threshold for evaluation;mAP^IoU=0.5:0.95^ was defined as the mean value of multiple mAPs that were evaluated based on different IoU thresholds ranging from 0.5 to 0.95 at intervals of 0.05.

The computational cost was evaluated by the number of floating-point operations required to process a single image. The model size of a network was evaluated by the number of parameters of the network.

### 4.2. Comparisons

In this section, we compare the proposed TC-YOLO network with some advanced mobile networks for underwater object detection, namely YOLOv3, YOLOX-tiny, RetinaNet, Faster-RCNN, and the original YOLOv5. For YOLOv3, the MobileNet was employed as the backbone. For YOLOX, the tiny version was selected because of its small channel numbers for mobile applications. For RetinaNet, the EfficientNet was used as the backbone. For Faster-RCNN, ResNet18 instead of ResNet50 was employed as the backbone for mobile applications. For YOLOv5, YOLOv5s was selected with CSPDarknet53 as the backbone. CLAHE image enhancement and the optimal transport label assignment were applied to the TC-YOLO model to implement the proposed approach. We evaluated the performance metrics of these networks on the RUIE2020 dataset, in which all the networks were trained with the same settings. Table 2 shows the comparisons of performance, computational cost, and model size.

Comparing the existing mobile detection networks, it was revealed that YOLOv5s provides state-of-the-art performance. YOLOv5s with CSPDarknet53 is larger and more advanced than YOLOv3 and YOLOX-tiny and has higher precision and less computational complexity than RetinaNet and Faster-RCNN. Developed from YOLOv5s, the proposed TC-YOLO is only 10% larger than YOLOv5s in size and remarkably surpasses the state-of-the-art in underwater detection tasks. The overall precision and recall increased by 3.2% and 5.7%, respectively. Specifically, the proposed approach can detect dense and small targets rather well: The mAP^IoU=0.5:0.95^ improved by 7.1%. These improvements were achieved without significantly sacrificing computational efficiency; the computational cost was only increased by 16%. The proposed approach achieves an excellent balance between prediction accuracy and computational complexity.

### 4.3. Ablation Experiments

Ablation experiments were carried out to analyze the contribution of each of the following improvements: image enhancement using CLAHE, Transformer block in the backbone, and CA block in the neck. Table 3 shows the results of ablation experiments, in which the given time is the average processing time per frame, including image enhancement and inference time.

Applying image enhancement using CLAHE caused a trivial drop in precisions but a significant boost in recall, because improved sharpness and dark details effectively reduced the number of false negatives. Applications of self-attention and coordinate attention mechanisms both offer significant overall improvement in underwater detection performance. Applying the Transformer block to the minimum feature map in the backbone provided roughly a 1–2% increase in precisions and a 2% rise in recall. Applying the CA blocks before the head provided significant improvement in precisions and recall of about 2–4%. Regarding computational cost, CLAHE image enhancement required the most computation, increasing processing time per frame by about 2.3 ms. Self-attention is known for being computationally expensive, and in our case, the Transformer block slowed the processing time by about 1 ms. The CA blocks applied in this study proved to be highly efficient, only increasing the processing time by about 0.2 ms. When all the proposed techniques were applied, the processing time per frame was still small, increasing from 16.2 ms to 19.7 ms, which is acceptable for real-time detection tasks. In our tests and studies, we noticed that the application of CLAHE image enhancement did not produce significant improvement when the lighting condition and water purity were great. Therefore, we recommend users deactivate CLAHE image enhancement for efficiency if underwater image quality is good.

Figure 6 shows an example of the detection of sea urchins. The proposed approach proved effective. The CLAHE algorithm sufficiently enriched the details of the originally dark targets. Small and obscured sea urchins were successfully detected from the complicated background.

## 5. Conclusions

In this paper, we proposed a new underwater object detection approach suitable for mobile deployments. Underwater object detection is difficult due to poor image quality, limited computational capacity, and underwater targets that are often small, dense, overlapped, and obscured. A new detection network, TC-YOLO, was developed by embedding the Transformer encoder and a coordinate attention module into YOLOv5. The CLAHE algorithm was used for underwater image enhancement, and the optimal transport assignment method was applied for label assignment in training. Combining these techniques, our proposed approach achieved state-of-the-art performance on the RUIE2020 dataset while remaining computationally efficient for real-time underwater detection tasks. In the ablation experiments, it was revealed that the application of the coordinate attention module to the end of the neck is a highly effective and efficient way to improve the performance of detection networks.

## Figures and Tables

**Figure 1 sensors-23-02567-f001:**
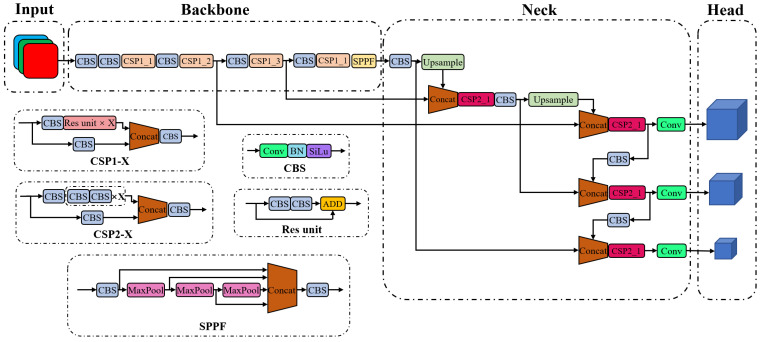
Original YOLOv5 network structure.

**Figure 2 sensors-23-02567-f002:**
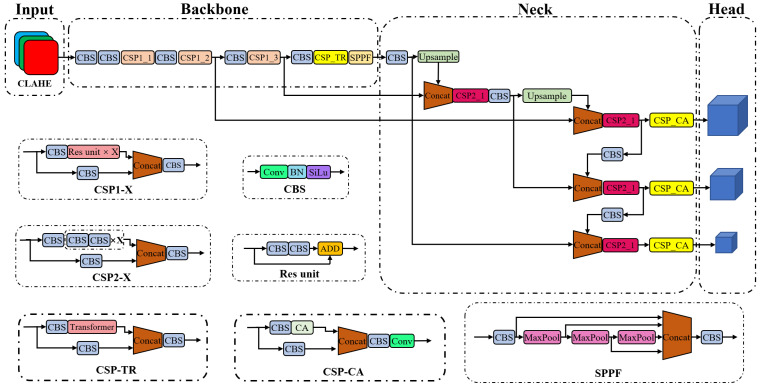
Overall structure of the proposed TC-YOLO network.

**Figure 3 sensors-23-02567-f003:**
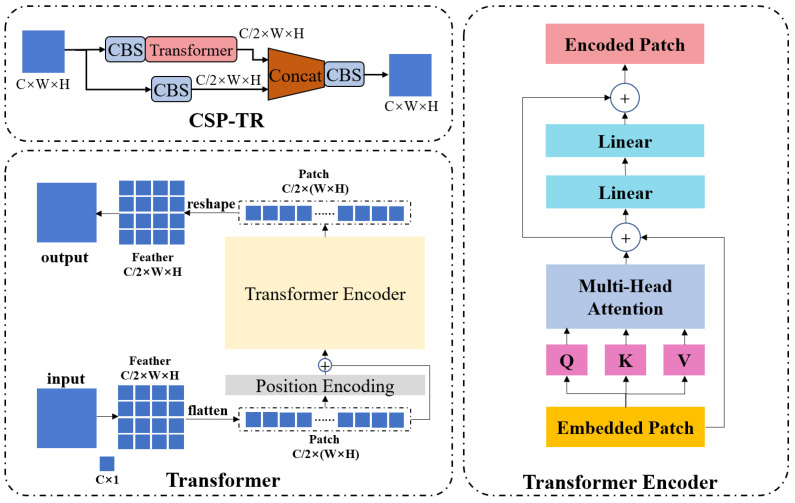
Detailed structure of the CSP-TR Block.

**Figure 4 sensors-23-02567-f004:**
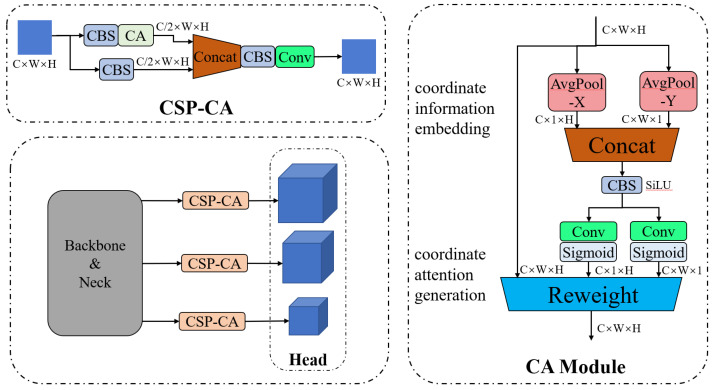
Structure of CSP-CA Block.

**Figure 5 sensors-23-02567-f005:**
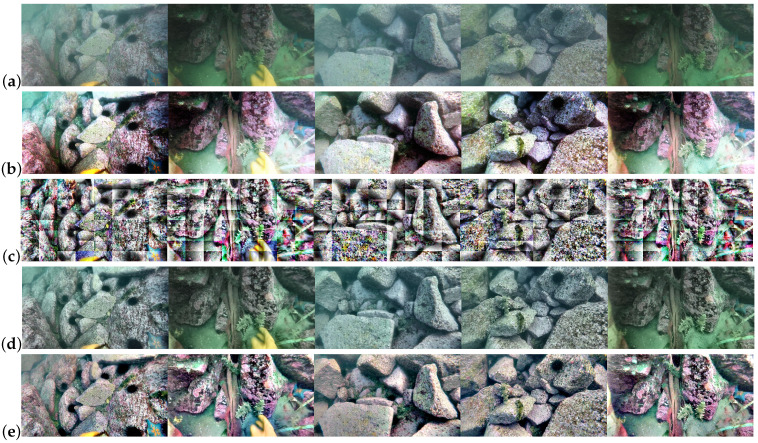
Comparison of underwater image enhancement results for different algorithms: (**a**) original images; (**b**) HE; (**c**) AHE; (**d**) CLAHE; (**e**) MSRCR.

**Figure 6 sensors-23-02567-f006:**
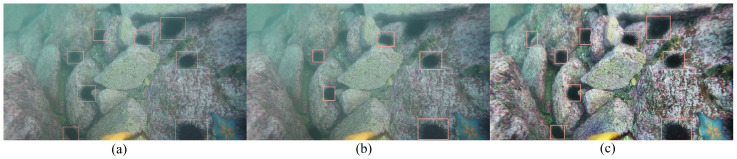
Demonstration of the proposed approach: GT of sea urchins (**a**); YOLOv5s prediction (**b**); TC-YOLO prediction (**c**).

**Table 1 sensors-23-02567-t001:** Comparison of detection performance using YOLOv5s for different image enhancement algorithms.

	Algorithms	Precision	Recall	Processing Time
1	Original Image	79.7%	71.1%	–
2	HE	77.1%	70.8%	3.2 ms
3	AHE	75.5%	68.4%	3.5 ms
4	CLAHE	78.4%	73.5%	3.3 ms
5	MSRCR	80.2%	74.3%	1646 ms

**Table 2 sensors-23-02567-t002:** Comparisons of mobile detection networks for underwater object detection.

	Model	Backbone	Precision	Recall	mAP^IoU=0.5^	mAP^IoU=0.5:0.95^	Floating-point Operations	Number of Parameters
1	YOLOv3 [11]	MobileNet [55]	70.6%	57.2%	70.2%	32.5%	6.58 $\times \;10^{9}$	4.5 $\times \;10^{6}$
2	YOLOX-tiny [56]	Darknet53	68.5%	59.8%	67.8%	34.6%	7.64 $\times \;10^{9}$	5.7 $\times \;10^{6}$
3	RetinaNet [57]	EfficientNet [58]	76.9%	63.6%	76.5%	40.7%	47.18 $\times \;10^{9}$	37.5 $\times \;10^{6}$
4	Faster-RCNN [59]	ResNet18	75.6%	65.1%	74.6%	41.9%	72.54 $\times \;10^{9}$	47.6 $\times \;10^{6}$
5	YOLOv5s (w/ OTA)	CSPDarknet53	79.7%	71.1%	76.5%	38.5%	16.00 $\times \;10^{9}$	7.0 $\times \;10^{6}$
6	TC-YOLO (w/ OTA & CLAHE)	CSPDarknet53	82.9%	76.8%	83.1%	45.6%	18.60 $\times \;10^{9}$	7.7 $\times \;10^{6}$

**Table 3 sensors-23-02567-t003:** Results of ablation experiments.

Case	CLAHE	Transformer	CA Block	Precision	Recall	mAP^IOU=0.5^	mAP^IOU=0.5:0.95^	Time
1	x	x	x	79.7%	71.1%	76.5%	38.5%	16.2 ms
2	*√*	x	x	78.1%	73.5%	75.4%	37.7%	18.7 ms
3	x	*√*	x	80.5%	71.8%	78.2%	40.3%	17.3 ms
4	x	x	*√*	81.2%	72.9%	78.6%	41.2%	16.5 ms
5	*√*	*√*	x	80.4%	74.2%	77.6%	39.5%	19.4 ms
6	*√*	x	*√*	80.9%	74.8%	79.6%	42.3%	18.8 ms
7	x	*√*	*√*	81.6%	75.1%	80.5%	43.5%	17.5 ms
8	*√*	*√*	*√*	82.9%	76.8%	83.1%	45.6%	19.7 ms

## Data Availability

Not applicable.

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
