# Peer review of "Underwater Object Detection Using TC-YOLO with Attention Mechanisms"

_sensors, 2023, doi:10.3390/s23052567_

Round 1
Reviewer 1 Report
Overall, the paper is well-structured and written. The reviewer has a few minor comments:
The author should add the structure of the paper at the end of the introduction section.
The previous work in the related work section should be linked with the proposed work.
Why the need for TC-YOLO?
Will the TC-YOLO pay any cost? Mean it will be always a win-win situation or there will be some situations, where its performance can be affected?
Reviewer 2 Report
Summary: This paper proposes to improve the performance of the YOLOv5 network for underwater object detection and small objects. The authors integrate the Coordinate Attention mechanism in their model and use Optimal Transport Assignment (OTA) to improve label assignment in training and employ the CLAHE algorithm for underwater image enhancement. With inspiration from several existing works, the authors have developed an object detection model.
The idea behind the paper is interesting and the improved performance seems reasonable. However, the proposed framework is presented poorly, i.e., the method section over explains the borrowed modules or parts from other works, such that the paragraphs and subsections are not following each other. The authors have to rewrite the method section to propose their framework cohesively. For example, lines 196-206 discuss to mention that the authors used a Transformer backbone, and the next paragraph starts by saying 'in this paper ...'
- The related work section fails to discuss works related to similar areas i.e., works that study underwater applications.
- Captions of the figures should be extended in a way that we could read the figure independently from the text. Full phrases of the abbreviations are missing.
- There are many abbreviations in the text that distracts the reader and makes reading very difficult.
Reviewer 3 Report
1 The dataset description should be moved to the materials and methods section with more details included.
2 The authors should have emploed cross-valiation for the evaluation method.
3 Are the reported results for validation? The evaluation method does not mentioned testing in data division.
4 How many anchors and what was the location of detection heads?
5 The choice of parameters and evaluation methods should be qualified with appropriate references, see similar studies utilizing Yolo can be cited so that to established the trustworthiness of the models and can provide reliabilit to baseline settings, see Detection of K-complexes in EEG signals using deep transfer learning and YOLOv3. Cluster Comput (2022). https://doi.org/10.1007/s10586-022-03802-0
6 why do you need object detection instead of classification?
7 The table of abbreviations is missing but required by the journal template.
Round 2
Reviewer 2 Report
The comments have been addressed at an acceptable level and the manuscript could be published in its current form.
Author Response
To the Reviewer and the Editor,
The authors would like to thank again the reviewer for the time and effort in reviewing this manuscript.
To address the comments from the Academic Editor, the authors have thoroughly polished the English writing of the manuscript. More related works published recently are added in Sections 2.1, 2.2, 2.3, and 2.4. Please see lines 78-86, 120-128, 171-182, and 194-208.
Best regards,
Dr. Shanran Tang